# Self-organization in brain tumors: How cell morphology and cell density influence glioma pattern formation

**Sara Jamous[1], Andrea Comba[2], Pedro R. Lowenstein[2]\*, Sebastien Motsch[1]\***

**1** Arizona State University, School of Mathematical & Statistical Sciences, Tempe, Arizona, United States of America, **2** University of Michigan, School of Medicine, Department of Neurosurgery and Rogel Cancer Center, Ann Arbor, Michigan, United States of America

\* pedrol@umich.edu (PRL); smotsch@asu.edu (SM)

**Data Availability Statement:** The source code used in the manuscript is available at https://figshare.com/articles/Code_Tumor_ellipse/12106800. It is written in Julia.

## Abstract

Modeling cancer cells is essential to better understand the dynamic nature of brain tumors and glioma cells, including their invasion of normal brain. Our goal is to study how the morphology of the glioma cell influences the formation of patterns of collective behavior such as flocks (cells moving in the same direction) or streams (cells moving in opposite direction) referred to as *oncostream*. We have observed experimentally that the presence of oncostreams correlates with tumor progression. We propose an original agent-based model that considers each cell as an ellipsoid. We show that stretching cells from round to ellipsoid increases stream formation. A systematic numerical investigation of the model was implemented in $\mathbb{R}^2$. We deduce a phase diagram identifying key regimes for the dynamics (e.g. formation of flocks, streams, scattering). Moreover, we study the effect of cellular density and show that, in contrast to classical models of flocking, increasing cellular density reduces the formation of flocks. We observe similar patterns in $\mathbb{R}^3$ with the noticeable difference that stream formation is more ubiquitous compared to flock formation.

## Author summary

Self-organization is the formation of large-scale multicellular patterns that result exclusively from the interactions amongst constituent single cells. To establish the existence of self-organization in brain tumors we used agent-based modeling based on data extracted from static and dynamic genetically engineered and implantable mouse glioma models. Implementation of our model in $\mathbb{R}^2$ identifies the dynamics that lead to formation of flocks (cells moving in a single direction), streams (cells moving in two directions), and cells moving as swarms or scattering. Increasing cellular density reduced formation of flocks and increased the formation of streams both in $\mathbb{R}^2$ and in $\mathbb{R}^3$. These results demonstrate the detailed mechanism leading to self-organization in brain tumors. As increasing density of oncostreams correlates with tumor malignancy, we establish a pathophysiological link between self-organization of glioma tumors and glioma malignancy. We propose the dismantling of oncostreams as a new therapeutic approach to the treatment of brain tumors.

**Funding:** PRL would like to acknowledge support from the National Institute of Health (NIH/NINDS Grants R01-NS076991 and R01-NS096756, NIH/NIBIB: R01-EB022563, NIH/NCI U01CA224160) and the Department of Neurosurgery and Leah's Happy Hearts. AC was funded in part by University of Michigan, MICHR Postdoctoral Translational 710 Scholars Program, TL1 TR002240-02, Project F049768. The funders had no role in study design, data collection and analysis, decision to publish, or preparation of the manuscript.

**Competing interests:** The authors have declared that no competing interests exist.

## Introduction

Primary brain tumors are one of the most lethal cancers. In spite of surgery, radiotherapy and chemotherapy, median survival remains at 14-18 months. The key to develop successful cancer therapy is to understand the essential mechanisms by which individual cancer cells proliferate, grow as a tumor, and invade normal brain. It is of particular importance to understand how individual cell morphology relates to collective macroscopic behaviors (e.g. stream formation, diffusion behavior). As our data indicate that glioma oncostreams promote tumor growth, this raises the question of whether cell morphology influences pattern formation and therefore the overall dynamics of growing tumors. This question is difficult to answer as it requires access to the time evolution of the positions of the cells *in vivo*.

We wished to explore the relationship of cell morphology to collective microscopic behavior patterns using mathematical modeling as it has already been successful in the exploration of various biologically relevant scenarios [1–4]. In particular, agent-based models (*modeling at the microscopic level*) are convenient as they incorporate essential features of cell behavior (i.e. motility, cell-cell interactions, etc.) and have been exploited to understand various self-organizing dynamical systems (e.g. pedestrians [5], birds [6], fish [7], bacteria [8, 9]).

To investigate the influence of the shape of the cell on tumor dynamics, we modeled cells as ellipses or ellipsoids. This assumption is motivated by experimental observations (see Fig 1) where cells within oncostreams display a length to width ratio of 2.7:1. Using ellipsoid shape is common in the study of bacteria, for instance viscoelastic ellipsoids have been used in [10] or self-propelled spheres in [11] (see also [12–17]). We were particularly interested in studying the dynamics in a regime of high cellular density where cells are always in contact with each other. 'Stretching' the cells' in this regime could potentially increase the formation of streams since streams would reduce overlapping of elongated cells. Indeed, in the context of soft-mater with elongated cylinders (e.g. nail, log, rice), stream formations are ubiquitous [18–20].

We propose an agent-based model that utilizes two mechanisms: i) self-propulsion, ii) cell-cell avoidance due to non-overlapping constraints. Since the cells have an ellipsoid shape, cell-cell avoidance leads to two possible effects: repulsion (i.e. cells move away from each other) and steering (cells turn to avoid collision). The larger the eccentricity of the cell, the larger the

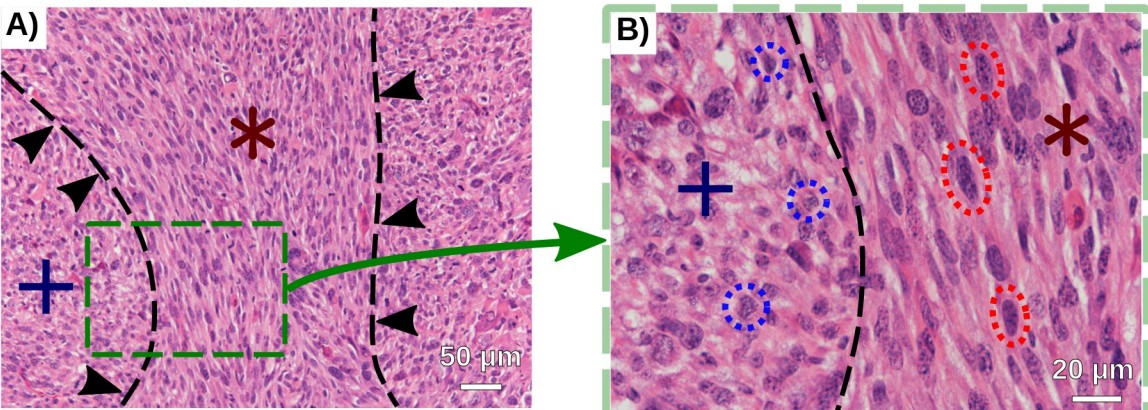

**Fig 1. Representative areas of elongated cells versus rounded cells found in glioma tumors.** Hematoxylin and eosin staining of a genetic engineered mouse glioma tumor expressing the following genetic lesions: Nras overexpression, shp53 downregulation, shATRx downregulation. **A)** Outlines and arrows demarcate multicellular structures formed by elongated cells (*) from areas of rounded cells (+). Scale bar 50*μm*. **B)** Image magnification of (**A**) highlighting in black broken lines the morphological differences between elongated (red) and rounded cells (blue). Scale bar 20*μm*.

effect on steering. In contrast to classical models of flocking [21, 22], in our model cells <u>do not</u> take into account the velocity of their neighbors.

We first investigated numerically in $\mathbb{R}^2$ how eccentricity influences flock formation (i.e. all the cells moving in the same direction) using as an indicator the polarization of the configuration. We observed that increasing eccentricity increases polarization. Surprisingly, this effect saturates and even becomes counterproductive as flock formation becomes less likely when eccentricity exceeds a threshold (eccentricity $e \approx .7$). Then, we studied how cellular density affects the dynamics by increasing the number of cells while maintaining the same size of the domain. Since we do not suppose a mean-field type interaction (there is no averaging in the interaction), increasing slightly the density could lead to drastic changes [23]. In our dynamics, we observed the emergence of streams when the density becomes large, meaning that cells are aligned but not necessarily moving in the same direction. We measure streams using the nematic average where we identify a vector $\omega$ and its opposite $-\omega$.

Beside the influence of cell morphology, the other key component of the dynamics is the strength of both the repulsive effect and the steering effect, as each determine the two coefficients $\alpha$ and $\beta$, respectively. One could speculate that increasing the steering effect (i.e. larger $\beta$) would enhance alignment and therefore lead to flocks or streams. Our numerical investigation revealed this not to be the case. Particularly at large densities, it is only when $\beta$ is small that a flock or a stream emerge. This result seems counter-intuitive. However, we need to emphasize that the alignment in our dynamics is only *indirect*, as cells do not perceive each other's velocity. Thus, it is an interplay between spatial constraint and steering that leads to the emergence of a stream or flock. Increasing a single parameter (repulsion or steering) does not necessarily enhance alignment.

Stream formation is more challenging to observe in $\mathbb{R}^3$ since cells avoiding each other no longer move aligned or in opposite direction as in $\mathbb{R}^2$. However, our simulations show that flock and stream formation do still occur in $\mathbb{R}^3$ providing that we maintain a large density of cells in the domain.

The complexity of the dynamics uncovered shows that it is difficult to predict *a priory* the effect of each mechanism. Therefore, it would be of great interest to develop a multi-scale approach to study the dynamics from a macroscopic viewpoint [24–27]. Moreover, this will facilitate data-model comparison [28, 29], as much of the experimental observations are made at a macroscopic scale. Investigating the partial-differential equation associated with the dynamics [30–32] could provide a way to bridge this gap.

The manuscript is organized as follows: we first present the agent-based model in section 1, then we study how the cell morphology influences the dynamics in section 1. A systematic numerical investigation of the model in $\mathbb{R}^2$ varying two key parameters is performed in section 1 which produces several phase diagrams of the dynamics at various densities. We explore the model in $\mathbb{R}^3$ in section 1 and draw our conclusions and future work in section 1.

## Material and methods

We propose an agent-based model to describe the motion of individual glioma cells. The dynamics combine cell-motility (i.e. self-propulsion) and cell-cell interaction (e.g repulsion or adhesion). Specifically, we consider $N$ cells described with a position vector $\mathbf{x}_i \in \mathbb{R}^d$ with $d$ the spatial dimension ($d$ = 2 or 3), moving with velocity $c\omega_i$ where $c > 0$ is the speed (supposed constant) and $\omega_i \in \mathbb{S}^{d-1}$ the velocity direction. The main novelty of the model is to consider an elliptic or ellipsoid shape for each cell. Thus, we consider two axes denoted $a$ and $b$ for (respectively) the major and minor axis (see Fig 2-left). As two cells cannot occupy the same spatial position, cells will *push each other* if they are too close. Thus, we define an interaction potential

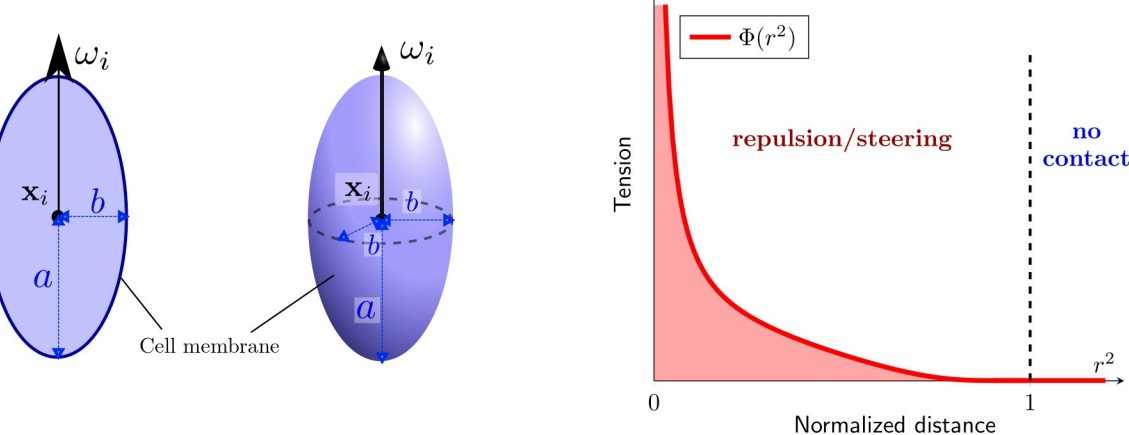

**Fig 2. Left**: a cell $i$ is described by its position $\mathbf{x}_i$, orientation $\omega_i$ and its elliptic shape determined by the two morphological components $a$ and $b$. **Right**: function $\Phi$ relies spacing between cell $r_{ij}$ (1) into *tension* that generates *repulsion* when two cells touch each other.

$V_i$ between cells that measures the *tension* exerted on cell $i$ generated by the surrounding cells:

$$V_i = \sum_{j=1, j \neq i}^{N} \Phi(r_{ij}^2) \quad \text{with} \qquad r_{ij}^2 = \left| \frac{\langle \mathbf{x}_j - \mathbf{x}_i, \omega_i \rangle}{a} \right|^2 + \left| \frac{\langle \mathbf{x}_j - \mathbf{x}_i, \omega_i^{\perp} \rangle}{b} \right|^2. \tag{1}$$

The quantity $r_{ij}$ is referred to as the *normalized distance* between the centers of the cells $i$ and $j$. For instance, if $a = b$ we recover that $r_{ij}$ is simply the norm $\|\mathbf{x}_j - \mathbf{x}_i\|/a$. The modification takes into account that the cell is more sensible to an obstacle in front rather an obstacle on the side. The model is defined in $\mathbb{R}^2$ (i.e. $d = 2$) and can be generalized to $\mathbb{R}^3$ by defining $r_{ij}$ as follows:

$$r_{ij}^2 = \frac{1}{b^2} \left( \| \mathbf{x}_j - \mathbf{x}_i \|^2 - e^2 [(\mathbf{x}_j - \mathbf{x}_i) \cdot \omega_i]^2 \right) \tag{2}$$

where $e \in (0, 1)$ is the eccentricity of an ellipse defined as $e = \sqrt{1 - \frac{b^2}{a^2}}$.

To prevent overlapping, the function $\Phi$ has to be singular at the origin. We choose the following smooth function with compact support (see Fig 2-right):

$$\Phi(s) = \begin{cases} \dfrac{1}{s} \exp\left(\dfrac{-1}{1-s}\right) & \text{if } 0 < s < 1 \\ \\ 0 & \text{if } s \geq 1. \end{cases} \tag{3}$$

As $r_{ij}$ decreases, $\Phi$ increases resulting into *repulsion*. We have now defined all the quantities required to define our agent-based model.

**Definition 1** *Consider* $(\mathbf{x}_i, \omega_i) \in \mathbb{R}^d \times \mathbb{S}^{d-1}$ *for* $i = 1..N$ *and the dimension space* $d = 2$ *or* $d = 3$*. The self-propelled dynamics are defined as*:

$$\dot{\mathbf{x}}_i = \overbrace{c\,\omega_i}^{\text{self-propulsion}} - \overbrace{\alpha \nabla_{\mathbf{x}_i} V_i}^{\text{repulsion}} \tag{4}$$

$$\dot{\omega}_i = \underbrace{-\beta P_{\omega_i^\perp}(\nabla_{\omega_i} V_i)}_{\text{steering}} \tag{5}$$

where *α*, *β* and *c* are positive constant, $V_i$ is the tension defined in (1) and $P_{\omega_i^\perp} = \mathrm{Id} - \omega_i \otimes \omega_i$ is the projector operator onto the the normal plane to *ω*$_i$ (it ensures that *ω*$_i$ stays of norm 1).

In order to reduce the tension generated by neighboring cells, a cell can either move away (i.e. *repulsion* effect) or change its direction (i.e. *steering* effect). Both maneuvers are pondered by the coefficients *α* and *β* representing the strength of each effect. Using the expression of $V_i$ (1), one can deduce an explicit expression of the dynamics (see S1 Text). Notice that if the cell has a circular shape (i.e. $a = b$ and the eccentricity $e = 0$), its orientation will remain constant i.e. $\dot{\omega}_i = 0$. Indeed, in that case, turning will have no effect on the reduction of the tension $V_i$. Thus, *steering effects* can only occur when if $a \neq b$.

**Remark 2** *Notice that* $r_{ij}$ *cannot be considered a distance between cells i and j as it is not symmetric (i.e.* $r_{ij} \neq r_{ji}$ *in general). Thus, the influence of the cell i on j is in general different from the cell j on i, i.e* $\Phi(r_{ij}^2) \neq \Phi(r_{ji}^2)$*.*

## Results

### Eccentricity effect on the dynamics

**Eccentricity induces alignment.**   Our first investigation of the agent-based model (4) and (5) is concerned with the impact of the morphology of the cell on the global behavior of the dynamics. As stated above, cells have perfect rounded shape when the two parameters *a* and *b* are equal (i.e. eccentricity *e* is zero) whereas they have elliptic or ellipsoid shape when $a > b$ (i.e. $e > 0$). We varied the eccentricity *e* and measured how this change affects the cells spatial configuration.

Before varying the morphological parameters *a* and *b*, there are several other parameters to be determined in our dynamics. When possible, we use experimental values that have been quantified in vivo. For instance, it has been observed that glioma cell size varies in between $5\mu m$ to $20\mu m$ for their diameter and their speed varies around $10\mu m/h$[33]. However, some parameters cannot be inferred from experimental observations such as the strength of the repulsion *α* and the steering *β*. A more detailed investigation of these two parameters will be conducted in the next section. For now, we fix their values as indicated in Table 1.

**Table 1. Parameters used for the simulations of Figs 3 and 5.**

| | | |
|---|---|---|
| Diameter cell (front/back) | **2a** | **8–14** $\mu m$ |
| Diameter cell (side) | **2b** | **6–8** $\mu m$ |
| Motility | $c$ | $10\mu m/h$ |
| Length domain Ω | $L$ | $300\mu m$ |
| Number of cells | $N$ | 1000 |
| Cell-cell repulsion | $\Phi$ | Eq (3) |
| Strength repulsion | $\alpha$ | $40\mu m^2/h$ |
| Strength steering | $\beta$ | $1h^{-1}$ |

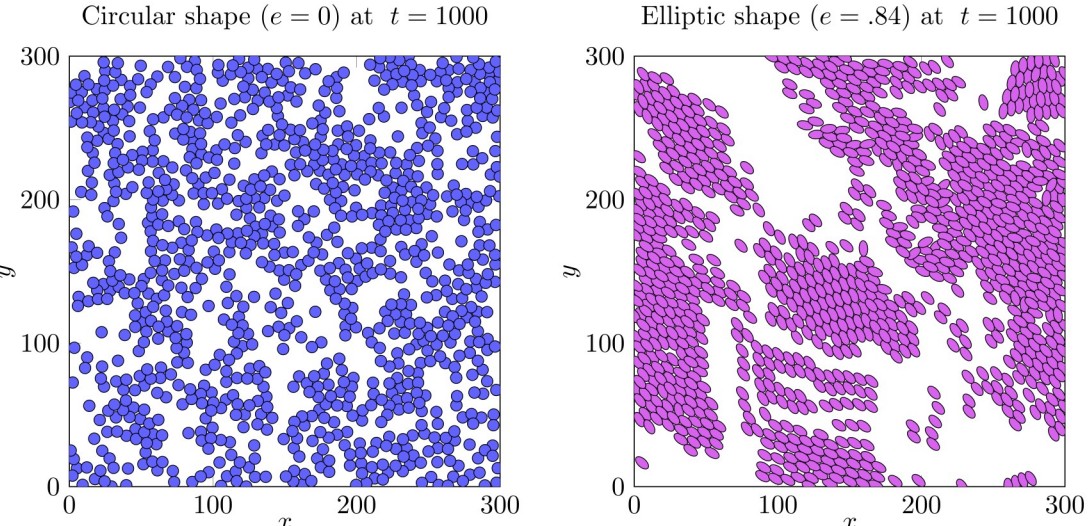

**Fig 3. Snapshot of the simulation of the model starting from a uniform distribution.** After $t$ = 1000 unit of time, circular cells ($a = b = 4\mu m$, $e = 0$) do not form any flocking pattern (**left**) whereas elliptic cells ($a = 5.5\mu m$, $b = 3\mu m$, $e = .84$) move in a common direction (**right**).

We perform the simulation in a square domain $\Omega = [0, L] \times [0, L]$ with periodic boundary conditions. For the initial condition, the positions of the $N$ particles $\{\mathbf{x}_i\}_{\{i = 1...N\}}$ are distributed uniformly $\Omega$ and their directions $\{\omega_i\}_{\{i = 1...N\}}$ are taken randomly on the unit circle $\mathbb{S}^1$. In Fig 3, we draw the final configuration of the dynamics after $T = 1000$ unit time for two types of cells: circular shape ($a = b = 4\mu m$, $e = 0$) and elliptic shape ($a = 5.5\mu m$, $b = 3\mu m$, $e = .84$). We observe that circular cells have no particular spatial organization (Fig 3-left) while elliptic cells have formed clusters moving in the same directions (Fig 3-right). The full simulation is also available (see S1 Video).

**Statistical characterization.** To further investigate the dynamics, we introduce several statistics to characterize the emergent behavior.

**Definition 3** *Consider a velocity distribution* $\{\omega_i\}_{i=1..N} \in \mathbb{S}^{d-1}$*. We denote by* $\psi$ *the polarization*:

$$\psi = \frac{1}{N}\left|\sum_{i=1}^{N}\omega_i\right|. \tag{6}$$

*Similarly, we define the nematic polarization* [12]:

$$\gamma = \sqrt{\left\langle\cos\left(2\theta_i\right)\right\rangle^2 + \left\langle\sin\left(2\theta_i\right)\right\rangle^2} \tag{7}$$

*where* $\theta_i$ *is the angle between the direction* $\omega_i$ *and the horizontal axis and* $\langle\rangle$ *denotes the averaging over the indices* $i$ *(i.e.* $\left\langle\cos\left(2\theta_i\right)\right\rangle = \frac{1}{N}\sum_{i=1}^{N}\cos\left(2\theta_i\right)$*).*

We define the configuration as a flocking configuration (i.e. cells moving in the same direction) when the polarization $\psi \approx 1$ and the nematic polarization $\gamma \approx 1$. We define the configuration as a streaming configuration (i.e. cells' directions are parallel but not necessarily moving in the same direction) when the nematic polarization $\gamma \approx 1$ and $\psi < 1$.

**Remark 4** *The nematic polarization can be generalized in higher dimensions (see* S1 Text*).*

The previous statistics only involve the velocity of the cells $\omega_i$. We propose a third statistics to also characterize the spatial configuration.

**Definition 5** *Consider a spatial configuration $\{\mathbf{x}_i\}_{i=1\ldots N} \subset \mathbb{R}^d$ and a fixed radius R. We say that cell i is linked to cell j if the distance between the two particles is less than R. This defines a relationship (i.e. $i \sim j$) with $i \neq j$ defined:*

$$i \sim j \quad if \ \ and \ \ only \ \ if \quad \parallel \mathbf{x}_j - \mathbf{x}_i \parallel \leq R. \tag{8}$$

*Clusters $\mathcal{C}_k$ are defined as the connected components for this relationship: two cells $i_0$ and $j_0$ belong to the same cluster if there exists particles $i_1, \ldots, i_k$ (a path) such that*

$$i_0 \sim i_1, \ \ i_1 \sim i_2, \ \ \cdots, \ \ i_k \sim j_0. \tag{9}$$

*The cluster size $|\mathcal{C}_k|$ denotes the number of cells belonging in the cluster k. The average cluster size $|\bar{\mathcal{C}}|$ is defined as the expected cluster size $|\mathcal{C}_k|$ over all the positions:*

$$|\bar{\mathcal{C}}| = \frac{1}{N}\sum_{i=1}^{N}|\mathcal{C}(\mathbf{x}_i)|, \tag{10}$$

*where $\mathcal{C}(\mathbf{x}_i)$ denotes the cluster containing the cell i.*

We illustrate the three statistics used in Fig 4.

In Fig 5—**left**, we measure the value of the polarization $\psi$ over time for different shapes of the cells by varying the coefficients *a* and *b*. When the cells have a circular shape ($a = b = 4\mu m$, $e = 0$), the polarization $\psi$ remains close to zero for all time whereas it increases up to a maximum close to 1 when the eccentricity is greater than zero. The relation between eccentricity and polarization is however non-trivial: increasing the eccentricity does not necessarily lead to large polarization. For instance, the polarization with eccentricity $e = .89$ is significantly smaller than with eccentricity $e = .84$.

To further investigate the relationship between polarization and eccentricity, we plot in Fig 5—**right** the polarization at the final time for several experiments (changing the seed for the initial condition) and various eccentricities *e*. We then perform a local regression ('loess' method) to estimate the expected polarization $\psi$ as a function of *e*. We observe that increasing the eccentricity *e* leads to larger polarization up to $e \approx .7$ but then the polarization quickly decays for larger eccentricities.

**Indirect alignment.** In classical models of flocking [21, 22], each individual has access to the velocity of its neighbors. By relaxing its own velocity toward the average velocity of its

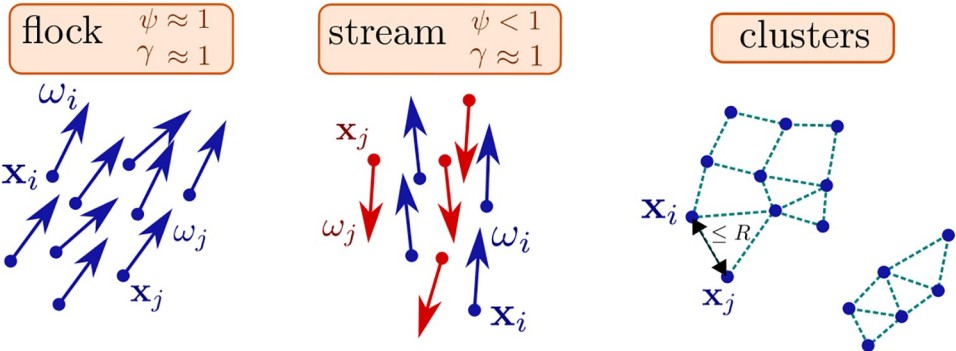

**Fig 4. The statistics used to characterize the dynamics: The polarization $\psi$ (6), the nematic polarization $\gamma$ (7) and the clustering (8) and (9).**

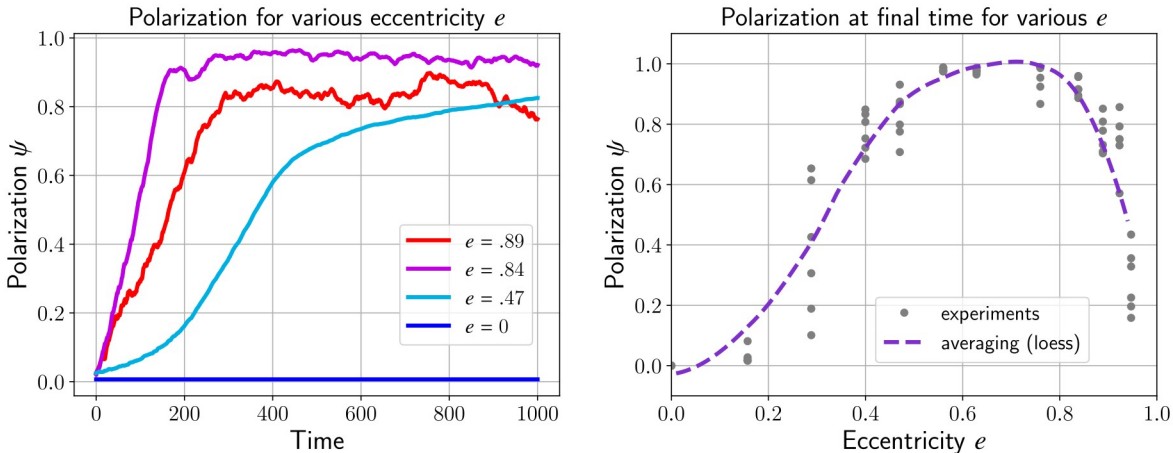

**Fig 5. Polarization $\psi$ (6) over time while varying the eccentricity of the cell $e$.** Ellipsoid cells that align will lead to an increase of $\psi$ close to its maximum 1. For the circular cells (blue curve), the polarization remains very low as no streams emerge from the dynamics (**left**). The polarization $\psi$ over eccentricity $e$ during the final time $t = 1000$ of the left figure will form a parabola. By increasing the eccentricity, there is no fundamental impact on the polarization coefficient of the cells (**right**).

neighbors, a flock emerges. This begs the question on how individual agents communicate. However, in the agent-based model we propose (4) and (5), the cell $i$ has no knowledge of the velocity of any of its neighbors, i.e. $\omega_j$ is not used to define the evolution of $(\mathbf{x}_i, \omega_i)$. Therefore, it is unclear at first why the dynamics proposed could generate similar flocking patterns.

To address this question, we provide a linear perturbation analysis of the model with respect to the eccentricity in the case of only two cells $i$ and $j$. Let's denote $\theta_{ij}$ the angle between the direction of the cell $\omega_i$ and the relative position vector $\mathbf{x}_j - \mathbf{x}_i$ (see Fig 6). Thus, one can

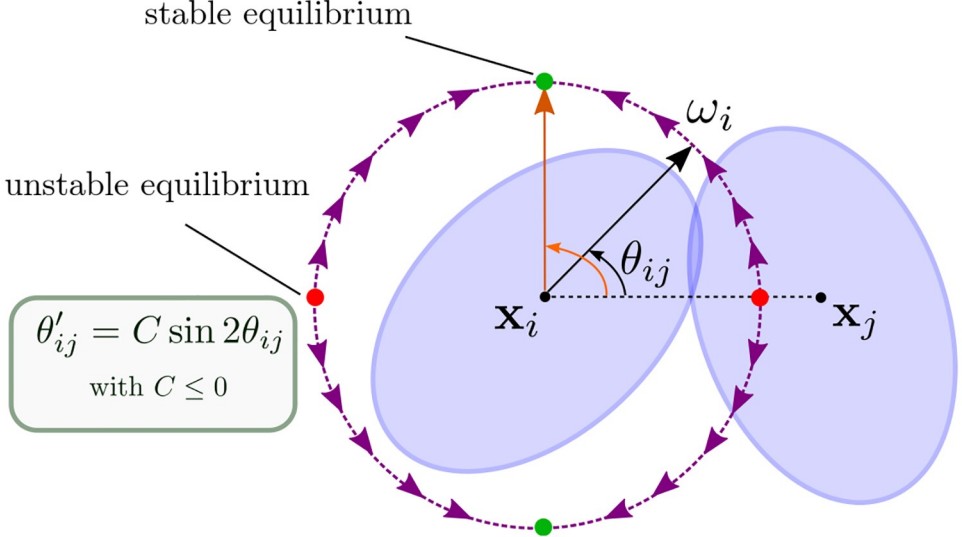

**Fig 6. Indirect alignment of two cells $i$ and $j$.** Both cell $i$ and $j$ will rotate to align with the orthogonal vector to $\mathbf{x}_j - \mathbf{x}_i$.

write $\omega_i = (\cos\theta_{ij}, \sin\theta_{ij})^T$ using the basis $\{\omega_i, \omega_i^\perp\}$. In particular,

$$\omega_i' = (-\sin\theta_{ij}, \cos\theta_{ij})^T \theta_{ij}' = \omega_i^\perp \theta_{ij}'.$$

We deduce that:

$$\theta_{ij}' = \beta \frac{2e^2}{b^2} \Phi'(r_{ij}^2) x_{ij} y_{ij}.$$

where $x_{ij} = \langle \mathbf{x}_j - \mathbf{x}_i, \omega_i \rangle$, $y_{ij} = \langle \mathbf{x}_j - \mathbf{x}_i, \omega_i^\perp \rangle$. Moreover, elementary geometry shows that $x_{ij} = |\mathbf{x}_j - \mathbf{x}_i| \cos\theta_{ij}$ and $y_{ij} = |\mathbf{x}_j - \mathbf{x}_i| \sin\theta_{ij}$ and thus:

$$\theta_{ij}' = C \sin 2\theta_{ij}, \qquad \text{with } C = \beta \frac{e^2}{b^2} \Phi'(r_{ij}^2) |\mathbf{x}_j - \mathbf{x}_i|^2. \qquad (11)$$

Notice that $C \leq 0$ since $\Phi'(r_{ij}^2) \leq 0$. As a consequence, if $i$ and $j$ stay close enough (e.g. $r_{ij}^2 < 1$), there are two stable equilibria for $\theta_{ij}$ at $\pm\pi/2$. Sketching the phase portrait in Fig 6 indicates that $\omega_i$ will rotate toward the stable equilibrium to align with $(\mathbf{x}_j - \mathbf{x}_i)^\perp$. By a similar argument, $\omega_j$ will be orthogonal to $(\mathbf{x}_j - \mathbf{x}_i)$ as well.

Therefore, instead of a direct alignment between $\omega_i$ and $\omega_j$, we have an indirect alignment: both vectors will align with $(\mathbf{x}_j - \mathbf{x}_i)^\perp$. Notice that this indirect form of alignment allows for the two vectors $\omega_i$ and $\omega_j$ to be *negatively* aligned, i.e. $\omega_i = -\omega_j$ which could lead to streaming formation. In dimension larger 2, $(\mathbf{x}_j - \mathbf{x}_i)^\perp$ is an hyperplane, thus even if $\omega_i$ and $\omega_j$ become orthogonal to $(\mathbf{x}_j - \mathbf{x}_i)^\perp$, it is insufficient to conclude that $\omega_i$ and $\omega_j$ will become parallel. Indeed, as we will show numerically in the next section, one has to consider also the spatial configuration (e.g. density) to predict whether the dynamics will generate flock or stream formations.

## Density effect

In the previous section, we investigated how cell morphology (i.e. $a$, $b$) promotes the emergence of flocking patterns (i.e. cells moving in the same direction). Our formal analyses show that we could also observe stream formation (i.e. cells moving in opposite directions). In this section, we will define the conditions under which streams emerge. To define the conditions which allow the emergence of streams we will study the dynamics of our system as we vary the parameters $\alpha$ (strength repulsion), $\beta$ (strength steering) and $N$ (density). We will fix the shape of the cells with $a = 5.5\mu m$ and $b = 3\mu m$ as they are the most common values experimentally. The range of the parameters are given in Table 2.

**Emergence of streams.** To illustrate the formation of streams (see Eqs (4) and (5)), we perform simulations within the parameter constraints of: $\alpha = 100$, $\beta = .1$ (strong repulsion, low

**Table 2. Parameters used for the simulations of Figs 7–13.**

| Diameter cell (front/back) | $2a$ | $11\mu m$ |
|---|---|---|
| Diameter cell (side) | $2b$ | $6\mu m$ |
| Motility | $c$ | $10\mu m/h$ |
| Length domain $\Omega$ | $L$ | $300\mu m$ |
| Number of cells | **N** | **1000 − 2000** |
| Cell-cell repulsion | $\Phi$ | Eq (3) |
| Repulsion strength | $\alpha$ | **10 − 200 $\mu m^2/h$** |
| Steering strength | $\beta$ | **.1 − 10 $h^{-1}$** |

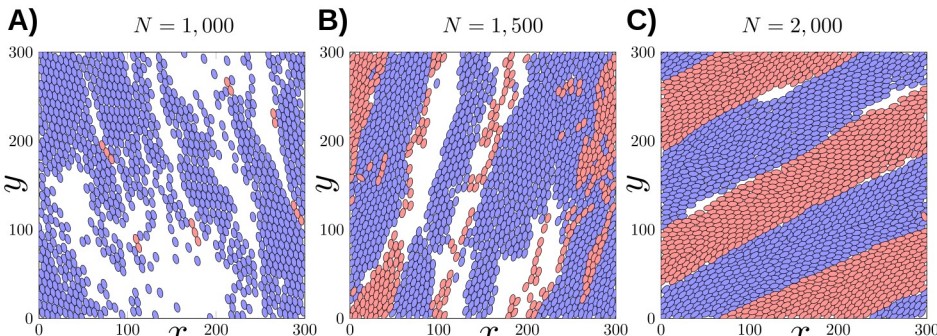

**Fig 7. Snapshots of the simulation of the dynamics at *t* = 1000 unit time for various cell densities (*N* = 1000, 1500 and 2000).** Red cells are moving in opposite direction to the blue cells. Flocking appears when the density is low (**A**) but then the dynamics start to converge to stream formation as we increase the density (**B**-**C**).

steering). In Fig 7, we illustrate snapshots of three simulations where we increase the density from *N* = 1000 to *N* = 2000 at *t* = 1000 unit time. Cells are color coded by orientation: we determine the nematic average direction $\Omega_{nem}$ (see S1 Text) and then color each cell *i* blue if $\langle \omega_i, \Omega_{nem} \rangle > 0$ or in red if $\langle \omega_i, \Omega_{nem} \rangle < 0$.

We notice that when the number of particles is low with *N* = 1000 (Fig 7a), almost all the cells are perfectly aligned in the same direction leading to a flocking configuration. The evolution of the polarization $\psi$ given in Fig 8 confirms this observation since $\psi$ becomes close to 1 for *N* = 1000. As we increase the density with *N* = 1500 (Fig 7b), we observe that the number of cells moving in the opposite direction becomes larger making the polarization decay to only $\psi$ = .2. Finally in the case where *N* = 2000 (Fig 7c), the number of cells moving in opposite direction becomes balanced and we observe the formation of a stream where the flow inside the domain is bidirectional. Indeed, the polarization $\psi$ is close to zero for *N* = 2000 where the nematic polarization $\gamma$ is around .9 (see S2 Video).

**Local minimum for the energy.** We conclude that increasing the density of cells is the underlying mechanism for stream formation. However, one has to notice that we always use as initial configuration random configurations for the velocities $\omega_i$. If one would start from a

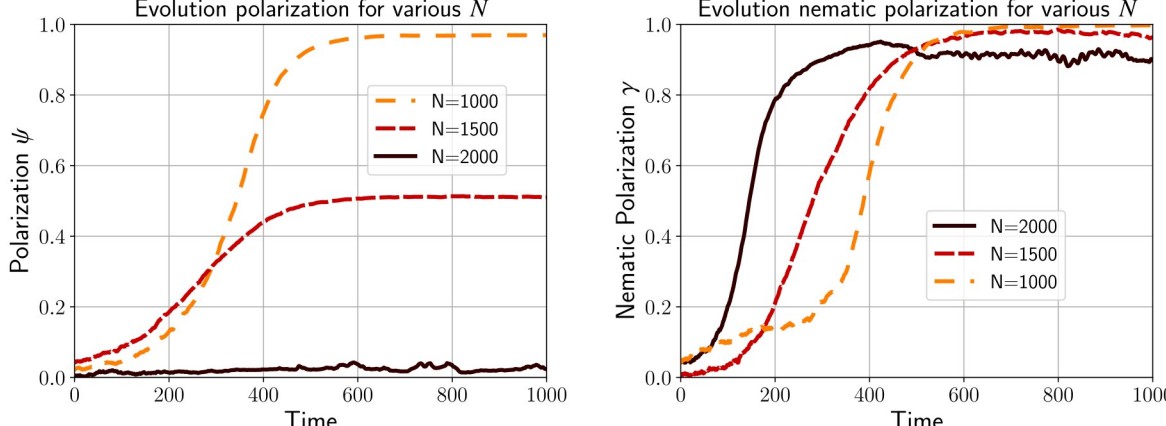

**Fig 8. Polarization $\psi$ and nematic polarization $\gamma$ for the simulations of Fig 7.** A flock occurs at low density (i.e. *N* = 1000) where $\psi$ and $\gamma$ converge approximately to 1, whereas streams emerge at larger density (i.e. *N* = 1500 and *N* = 2000).

perfect flock with no overlapping (i.e. $\omega_i = \Omega$ for all $i$ and $r_{ij} > 1$ for all $i, j$), then the configuration will remain in this configuration, it will be simply transported with a constant velocity. Thus, we will not observe the formation of a stream even at large cell density (e.g. $N = 2000$). In other words, the flocking configuration can be a globally stable configuration. In contrast, in a stream formation, the cells at the border between regions moving in opposite direction are in an unstable equilibrium (see Fig 6). Therefore, if the steering coefficient $\beta$ remains small enough, the streaming configuration will be maintained, the non-overlapping physical constraint (through the repulsion $\alpha$) prevents the cells from turning.

Another formal justification for the emergence of the stream configuration comes from the total potential energy $V$ (1):

$$V = \sum_{i=1}^{N} V_i = \sum_{i,j=1,\, j \neq i}^{N} \Phi(r_{ij}^2), \tag{12}$$

with $r_{ij}$ given by (1). The dynamics (4) and (5) is a gradient descent of the potential $V$ (i.e. $\mathbf{x}_i' = -\alpha \nabla_{\mathbf{x}_i} V_i$, $\omega_i' = -\beta P(\nabla_{\omega_i} V_i)$) combined with a free transport component (i.e. $\mathbf{x}_i' = c\omega_i$). The gradient descent decays the potential $V$ whereas the free transport could either increase or decrease $V$. But as we increase $\alpha$, the dynamics become more likely to become fixed in a local equilibrium (i.e. stream). Perturbations to the free transport component of the dynamics will be insufficient to move the configuration away from a local equilibrium (see Fig 9). However, on a large time scale, it is still possible that a stream configuration would eventually become a flock. The reverse situation, a flock becoming a stream, is unlikely as it would require an increase in the potential energy $V$. Since the dynamics is not conservative, we cannot rule out this scenario but numerically we haven't observed such transition.

**Phase diagram.** We have identified two configurations: flocking when the cells are aligned (i.e. $\psi \approx 1$, $\gamma \approx 1$), stream when the cells are moving in opposite directions (i.e. $\gamma \approx 1$). The convergence of the dynamics toward one of these configurations depends on the density $N$

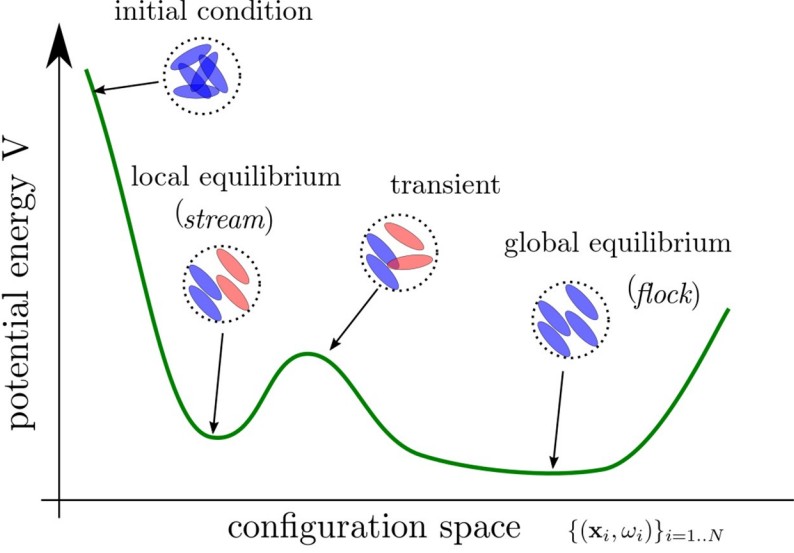

**Fig 9. Sketch representation of the potential energy $V$ (12) over the configuration space $\{(\mathbf{x}_i, \omega_i)\}_{i=1,\ldots,N}$.** Stream can be seen as local equilibrium whereas flock are global equilibrium. When the parameter $\alpha$ (repulsion) is increased, the stream configuration become more stable.

(Figs 7 and 8) and on the parameters $\alpha$ and $\beta$. We would like to study the effects of repulsion and alignment (i.e the coefficients $\alpha$ and $\beta$ resp.) for the three distinct cases of $N$.

With this aim, we fix the shape of the cells ($a = 5.5\mu m$ and $b = 3\mu m$) and make a systematic analysis by varying continuously $\alpha \in [0, 200]$ and $\beta \in [0, 10]$. For each value of $\alpha$ and $\beta$, we perform 5 simulations and compute several statistics after $t = 1000$ unit of time. For instance, in Fig 10 (top-left) we plot the polarization $\psi$ depending on $\alpha$ and $\beta$ in the case $N = 1000$. The scatter plot represents all the data points $(\alpha_j, \beta_j, \psi_j)$. Notice that for a given set $(\alpha, \beta)$ we find varying polarization $\psi$ due to random initial conditions. We represent the average polarization $\langle \psi \rangle$ as a surface computed from the 5 final configurations with similar parameters $(\alpha, \beta)$. To reduce the fluctuation, we estimate a local averaging ('loess') in Fig 10 (top-right) which makes possible the estimation of a smooth region where the polarization $\psi$ is higher than a given threshold. Regarding the use of averaging or smoothing, we observe that the polarization is surprisingly small when $\beta$ is large and $\alpha$ small. There is also another region where the polarization decays when $\alpha$ is large and $\beta$ is small.

A better visualization is to draw the polarization $\psi$ as a heat-map depending on $\alpha$ and $\beta$ (Fig 10 center-left). Through the use of smoothing, we can also estimate contours at different thresholds ($\psi = .5$ and $\psi = .8$). We proceed similarly with the nematic polarization $\gamma$ (7) in Fig 10 center-right. We notice that in contrast to the polarization $\psi$, the nematic polarization remains large even when $\beta$ is small and $\alpha$ is large: indeed, this is the regime where we observe the formation of streams.

Finally, we also use a third statistic to characterize the configuration using the average cluster size $|\bar{\mathcal{C}}|$ (10). We use the radius $R = 10\mu m$ to define the clusters (i.e. two cells are connected if their distance $\|\mathbf{x}_j - \mathbf{x}_i\|$ is less than $10\mu m$). The average size cluster $|\bar{\mathcal{C}}|$ is then estimated in Fig 10 (bottom). We observe two regions: cluster sizes are (relatively) smaller when $\alpha$ is small and independent of $\beta$. Thus, the repulsion $\alpha$ governs the formation of clusters.

We combine the three statistics (polarization $\psi$, nematic polarization $\gamma$, cluster size) to create a phase diagram in the parameter space $(\alpha, \beta)$. Three regions are delimited:

i). **flocking**: {$a$, $b$ such that $\psi > .8$},

ii). **streaming**: {$a$, $b$ such that $\gamma > .7$ and $\psi < .8$},

iii). **scattering**: {$a$, $b$ such that $|\bar{\mathcal{C}}| < 600$}.

The results are given in Fig 11. For most of the parameters $\alpha$ and $\beta$, the dynamics converge to a flock.

Performing a similar investigation for $N = 1500$ and $N = 2000$ lead to drastically different results. The regions where flocking occurs are more narrow (Fig 12a). But surprisingly stream formation is still occurring for all values of $\alpha$ as long as the steering coefficient $\beta$ is small enough (Fig 12b). Only the cluster formation through the statistic $|\bar{\mathcal{C}}|$ remains similar (see Fig 12c) as in the case $N = 1000$. As a result, the phase diagrams for $N = 1500$ and $N = 2000$ contain a large region not identifiable as either flock or stream (Fig 13). Notice that increasing density does not penalize the formation of streams in the region where $\beta$ is small and $\alpha$ is large.

## Dynamics in 3D

Finally, we would like to study the dynamics (4) and (5) in $\mathbb{R}^3$. There are several key differences between $\mathbb{R}^2$ and $\mathbb{R}^3$ for the dynamics. Our formal discussion in see section 1 showed that the dynamics enforce that nearby cells (denoted $i$ and $j$) must have their velocity ($\omega_i$ and $\omega_j$) orthogonal to their relative position ($\mathbf{x}_j - \mathbf{x}_i$). In $\mathbb{R}^2$, we concluded that nearby cells must be aligned at equilibrium, i.e. $\omega_i = \omega_j$ or $\omega_i = -\omega_j$. This is no longer the case in $\mathbb{R}^3$: $\omega_i$ and $\omega_j$ could

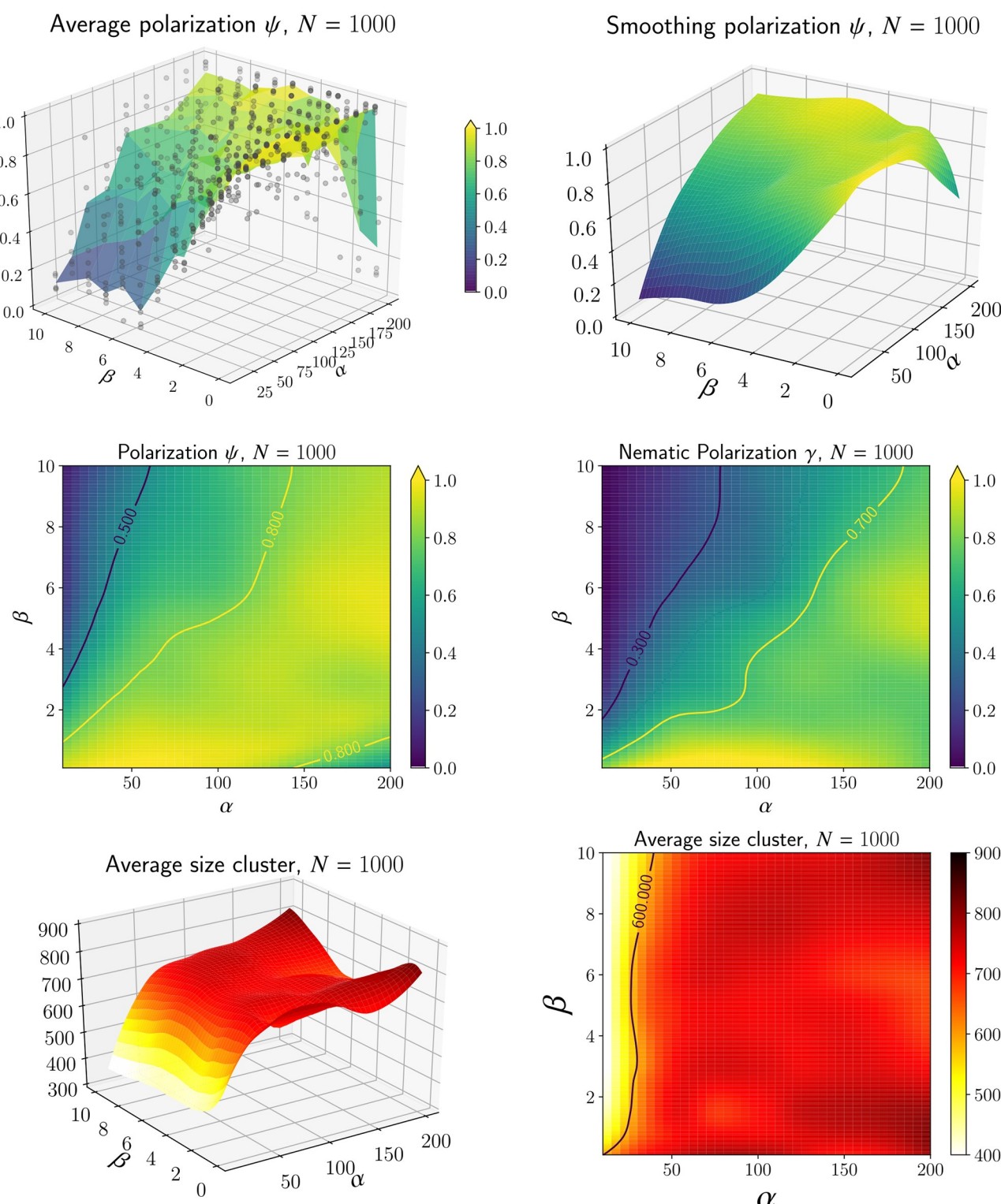

**Fig 10. Top**-left: for each pair $\alpha$, $\beta$, we estimate the polarization $\psi$ (scatter point) at the end of 5 simulations. The average is then computed to construct a surface plot. **Top**-right: we use a local regression ('loess') to estimate $\psi$ as a function of $\alpha$ and $\beta$ reducing the fluctuation. **Center**-left: heat-map representation of the polarization $\psi$ as a function of $a$, $b$ using the smooth estimation of $\psi$. The contour $\psi = .8$ will be used to determine the region when the dynamics generate flock. **Center**-right: we perform a similar analysis as the left figure using the nematic polarization $\gamma$. **Bottom**: the average size cluster $|\bar{\mathcal{C}}|$ (10) for various values of $\alpha$ and $\beta$. The estimation has been smoothed using local regression ('loess'). We then deduce the region when the cluster size is below a certain threshold.

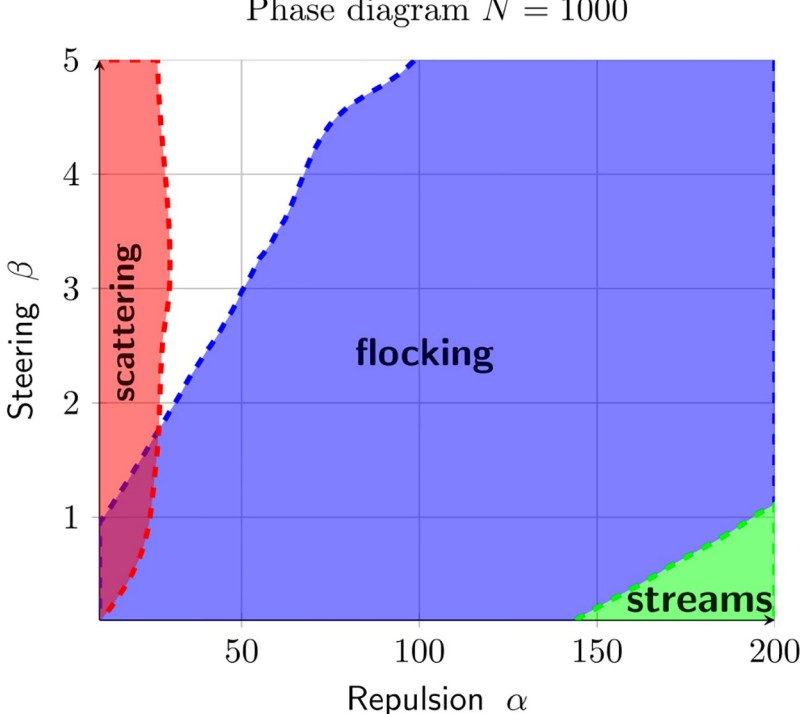

**Fig 11. Combining the results of Fig 10, we create a phase diagram consisting of three regions for the configuration: Flocking ($\psi > .8$), streams ($\gamma > .7$ and $\psi < .8$) and scattering ($|\bar{\mathcal{C}}| < 600$).**

be orthogonal to $(\mathbf{x}_j - \mathbf{x}_i)$ *without* being aligned. Therefore, it is uncertain if one should observe the emergence of flock or stream formations in $\mathbb{R}^3$.

Another difference in $\mathbb{R}^3$ is that the nematic polarization $\gamma$ is no longer defined as a velocity vector $\omega$ in $\mathbb{S}^2$ is defined using two angles instead of one. However, we have provided an alternative quantity denoted $J$ in S1 Text. We also use a smaller domain for our simulation in order to keep the same density as in $\mathbb{R}^2$ (maintaining a similar ratio "volume occupied/volume domain"); thus we reduce the size of the box to $L = 70\mu m$. Otherwise, the other parameters remain those of the previous simulations (see Table 3), in particular we use $\alpha = 100$ and $\beta = .1$ to be in the regime more susceptible of stream formation (at least in $\mathbb{R}^2$).

First, we investigate the model with $N = 1000$ cells (low density). We plot the final configuration at $t = 1000$ unit times starting from two different initial conditions in Fig 14. As in Fig 7, we color code the cells depending on their orientation to help visualize cells moving in opposite directions. Notice that cells do not necessarily move parallel to each other (they can move orthogonal to each other). But after a transient period, only one or two directions remain as cells form either a flock or a stream. Indeed, we observe the formation of a flock (Fig 14-top left) and of a stream (Fig 14-top right). Note that even when the flock develops (top **left**), few isolated cells (red) are still moving in opposing direction to to the main flow (blue). Thus, flock and stream configurations can emerge when the cell density is low.

The situation is different when we increase the density to $N = 1500$ and $N = 2000$. In this case we only observe the formation of streams (see Fig 14-bottom). Similar to the situation in $\mathbb{R}^2$, increasing the density reduces the possibility for the cells to rotate and therefore streams are more likely to occur. Plotting the time evolution of both the polarization $\psi$ and nematic

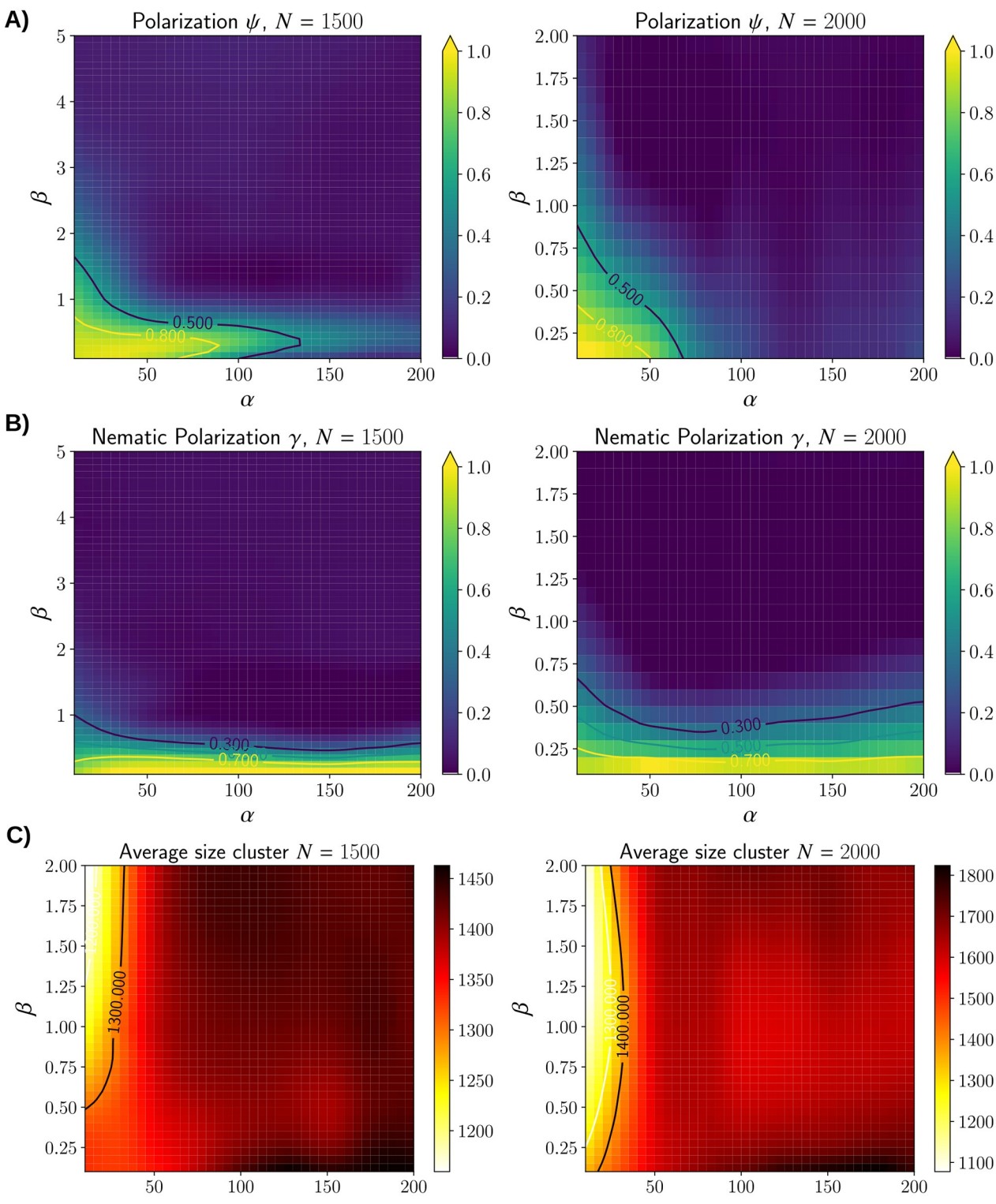

**Fig 12. A**) Average polarization $\psi$ for various parameters $\alpha$ and $\beta$ with $N = 1500$ and $N = 2000$. Notice that the polarization is significantly smaller compared to the case $N = 1000$ (Fig 10). **B**) Nematic polarization $\gamma$ for $N = 1500$ and $N = 2000$. $\gamma$ remains close to 1 for any values of $\alpha$ when $\beta$ is small. **C**) The average cluster size $|\bar{\mathcal{C}}|$ behave similarly as in the case $N = 1000$ with smaller clustering for small value of $\alpha$.

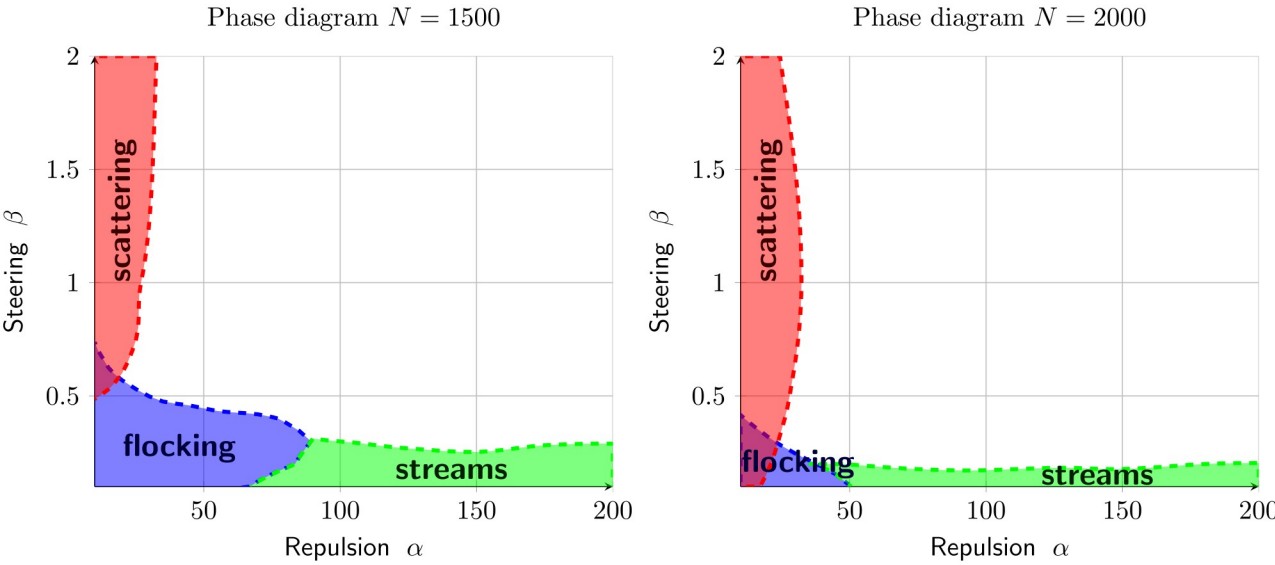

**Fig 13. Phase diagram when the total number of cells $N$ is 1500 (left) and 2000 (right).** As we increase the density, the regions for flocking configurations drastically reduce. However, streams are still form when $\beta$ is small.

polarization $J$ in Fig 15 confirms our observations. The quantity $J$ always converges to 1 whereas the nematic polarization $\psi$ stays low except when $N = 1000$.

## Discussion

Whether self-organization exists in brain tumors is incompletely understood. The identification of multicellular structures in brain tumors suggest that single cells are able to coalesce into multicellular patterns, possibly as a result of self-organization. We have recently described such multicellular structures within gliomas [34]. As these structures are reminiscent of streams described in other systems, we have labeled these structures oncostreams. Oncostreams extend over $100–500\mu m$ long and $50–200\mu m$ wide, and contain elongated cells. In this work we aimed to understand the role of the elongated morphology of cells within such oncostreams in the formation of the multicellular patterns, and whether these patterns are the result of self-organization operating within gliomas.

We propose an agent-based model that describes the motion of cancer cells and the emergence of pattern formation within gliomas. In our model, the morphology of the cells plays a key role in glioma pattern formation since cell eccentricity allows the cells to align (indirectly)

**Table 3. Parameters used for the simulations in $\mathbb{R}^3$ (Figs 14 and 15).**

| Diameter cell (front/back) | $a$ | $11\mu m$ |
|---|---|---|
| Diameter cell (side) | $b$ | $6\mu m$ |
| Motility | $c$ | $10\mu m/h$ |
| Length domain $\Omega$ | $L$ | $300\mu m$ |
| Number of cells | **N** | **1000 − 2000** |
| Cell-cell repulsion | $\Phi$ | Eq (3) |
| Repulsion strength | $\alpha$ | $100\mu m^2/h$ |
| Steering strength | $\beta$ | $.1h^{-1}$ |

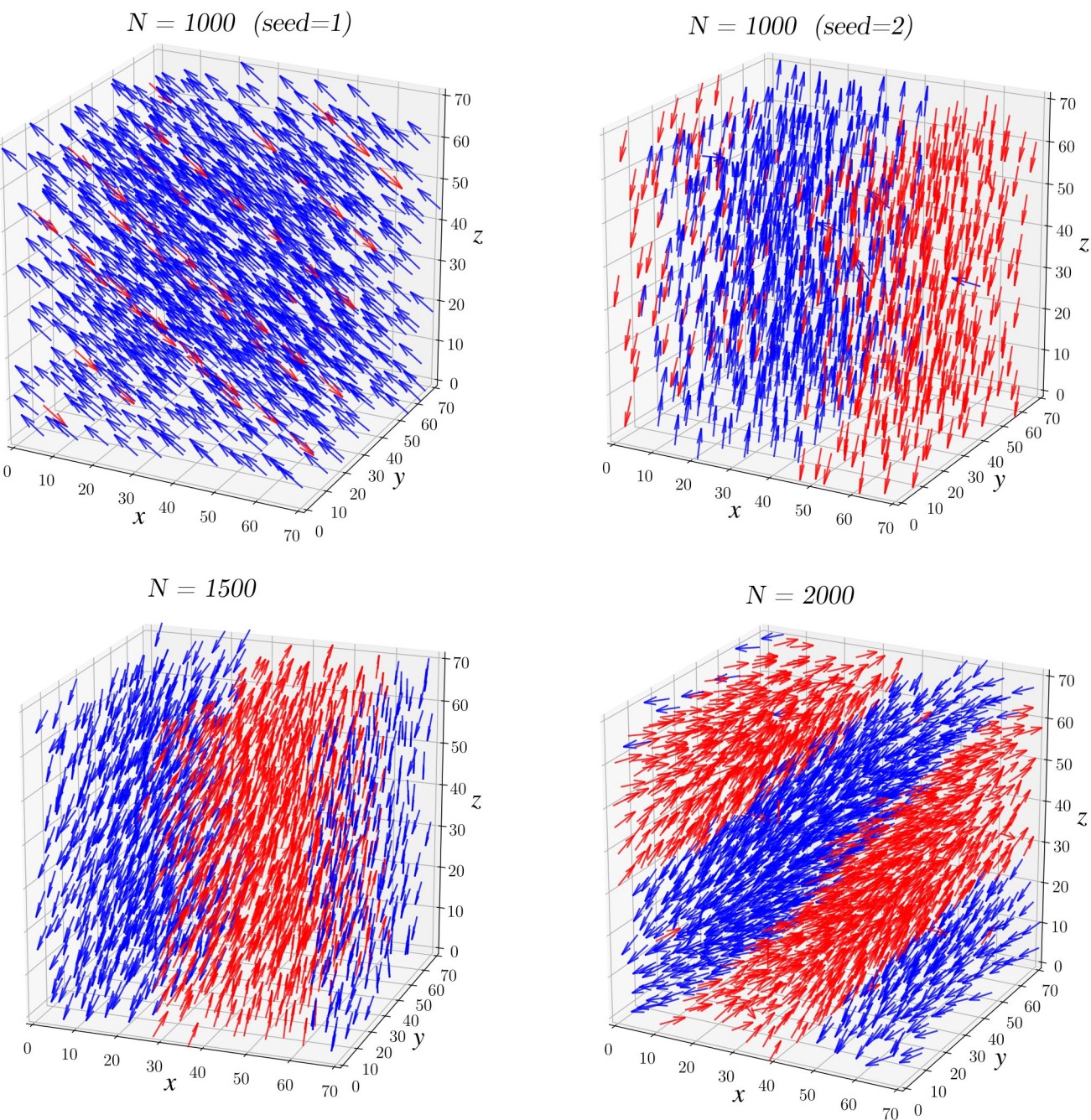

**Fig 14. Snapshots of simulation in $\mathbb{R}^3$ at $t$ = 1000 unit time with number of particles $N$ equal to 1000 (top), 1500 and 2000 (bottom).** Flocking and streaming appear when the number of particles is low depending on the initial condition (top **left** and top **right** respectively). Whereas only stream emerges when the number of particles is higher, N equal 1500 and 2000 (bottom—**left** and **right** respectively). We color code the cells in blue or red depending on the direction in comparison to the nematic average (see S1 Text).

to each other and eventually coalesce to form a flock or a stream. In the special case where cells are circular, cells cannot align and thus no flock or stream can formed. The emergence of such multicellular patterns is also governed by additional parameters, i.e. $\alpha$ (repulsion), $\beta$ (steering), and cell density. Several phase diagrams summarizing the effects of these parameters have

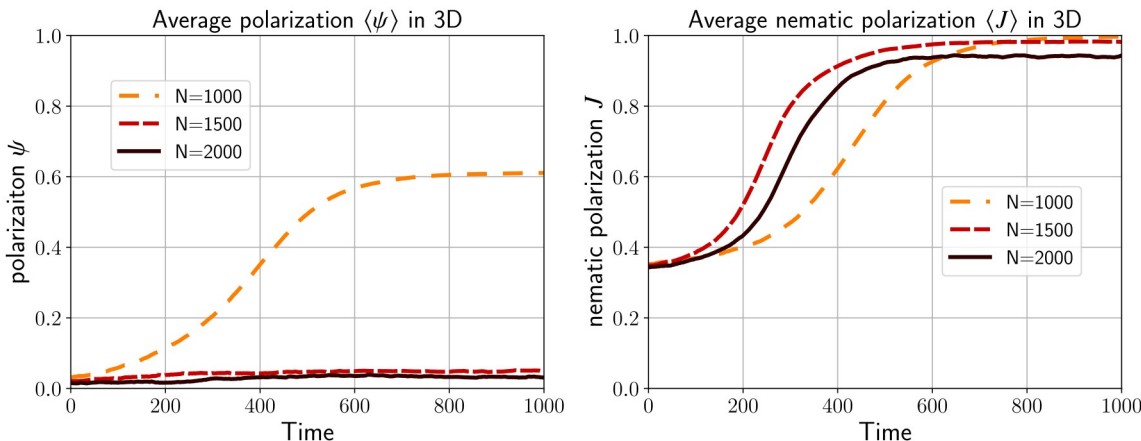

**Fig 15. The evolution of the polarization $\psi$ and the nematic polarization $J$ for the simulations presented in Fig 14.** When the number of particle is low ($N = 1000$), flock and stream emerge depending on the initial condition leading to an average value for $\psi$ is around 0.6 and $J$ close to 1. However, for larger density ($N = 1500$ and $N = 2000$), only streams emerge since the low polarization $\psi$ is low and the nematic polarization $J$ is close to 1.

been estimated for various densities. In contrast to mean-field type models, the density drastically changed the dynamics. Flocking configurations became more sparse and streams' density increased as cellular density increased. This has important biological implications as the density of gliomas is constantly changing.

We have discovered that glioma oncostreams indeed display characteristics of self-organization. Namely, our data strongly suggest that the emergence of the large-scale structures within brain tumors, flocks and streams, result from intercellular interactions. As the density of flocks and streams correlates with glioma aggressiveness, we can link the macroscopic behavior of brain tumors to the intercellular interactions of individual glioma cells. The formation of large scale patterns that result from intercellular interactions, and the fact that these structures determine glioma behavior, i.e., growth, invasion, aggressiveness, allow us to conclude that gliomas display clear evidence of self-organization, and that tumor self-organization plays an important role in tumor malignity.

Most of our experimental work was performed using mouse tissues and genetically engineered mouse models of brain tumors. However, the recognition of oncostreams in human malignant gliomas [35] strongly suggests that human brain tumors can also self-organize into structures that influence tumor malignity. We suggest that a novel approach to the treatment of brain tumors was to disassemble the oncostreams. Ongoing molecular studies of oncostreams indicate that this is feasible. Our data make important biological predictions.

Our model achieves flocking through cells of an eccentricity larger than 1, and parameters which determine cell repulsion, and cell steering. No function of cell adhesion is included into the model. This strongly suggests that the molecular intercellular interactions leading to stream and flock formation may regulate the degree of intercellular adhesion. It will be important to investigate the molecular basis of stream and flock formation in gliomas. The cytoskeleton is also likely to play a central role in stream and flock formation as the eccentricity has an optimal value to form oncostreams, and that, once exceeded, the capacity to form oncostreams decreases. Molecular mechanisms that might optimize cell eccentricity are currently not understood, and might yield important results concerning the molecular mechanisms that are necessary to form oncostreams and flocks.

Our data also suggest that intra-glioma dynamics are cell density dependent, as the directionality of the cells changes significantly upon increases of cell density. As intraglioma dynamics are important to tumor growth and invasion, changes achieved by therapeutic cytotoxic agents, may not eliminate glioma growth, but rather alter its organization and directionality. As methods become available to affect the overall organization of gliomas, we suggest that the effects on intratumoral dynamics be considered in terms of drugs' mechanisms of action. Increasing density is also likely to affect the overall macroscopic tumor growth, as flocks (which move in one direction), are able to convert into streams (which move in two directions). As a consequence, the quality and distribution of tumor growth may change in response to partial cytotoxicity to a diffuse tumor capable of growing in more directions as it invades surrounding normal brain.

The phase diagrams indicate the potential existence of critical points and phase transitions at which non-aligned cells can become aligned and form a flock or a stream, a relationship which is also dependent on glioma cell density. The existence of critical points will be important as they might regulate the sudden scattering of cells, or their organization into patterns of collective motion that are likely to determine tumor growth.

Our work proposes a number of extensions which will be pursued in the future. For instance, it will be important to mix cells with different shapes (i.e., with different values for $\alpha$ and $\beta$) since not all the cells are identical (see for instance [36]). We will also study how cell eccentricity varies over time and even whether it influences cell mitosis, and/or the birth/death cycles. Increasing the density is also challenging numerically as the dynamics become singular when two cells overlap which is more likely with a birth process. To avoid this complication, one could explore a continuous description of the dynamics through a Partial Differential Equation (PDE) [30–32]. Such PDE description might provide some hindsight about the emergence of flock or stream in certain regimes (e.g. $\alpha \gg 1, \beta \ll 1$).

Adding cell divisions will also raise new questions such as how fast do cell spread depending on their distance to the center of the tumor. It is yet to be determined whether cells will move faster close to the center (large density) or at the periphery of the tumor (low density).

Another extension of the model would consist in adding a "contact inhibition of locomotion" (CIL) to the cells [37]. The main idea is that cells would reduce their self-propulsion as they experience contact. As a result, we would expect that the perturbation due to the free transport component (see Fig 9) would be reduced and therefore flocks and streams would be more likely to occur at larger density. Such behavior would be consistent with experimental observations where 2D-cell layer clusters have been observed at large density [38].

In summary, through a detailed investigation of patterns of glioma growth and agent-based mathematical modeling we explain the importance of cell shape during glioma growth, and its consequences for glioma self-organization, aggressiveness and invasion. The long term consequences of glioma self-organization will impact our understanding of glioma biology, and suggest novel treatments.

## Supporting information

**S1 Text. Explicit expression of the model and definition of nematic average.**
(PDF)

**S1 Video. Circles VS ellipses.** Numerical simulation comparing the dynamics with circles (i.e. $a = b = 4\mu m$) and ellipses ($a = 5.5\mu m$, $b = 3\mu m$). Fig 3 corresponds to a screenshot of this video taken at $t = 1000$ unit time.
(AVI)

**S2 Video. Stream formation.** At large density, stream formations are more likely to occur (see also Fig 7). After a transient time, the dynamics generate streams that move in opposite direction. Cells have been colored depending on their direction: blue cells are the one moving to the right, red cells move left and white cells move up or down.
(AVI)

## Author Contributions

**Conceptualization:** Pedro R. Lowenstein, Sebastien Motsch.

**Data curation:** Andrea Comba, Sebastien Motsch.

**Investigation:** Sara Jamous, Andrea Comba, Pedro R. Lowenstein, Sebastien Motsch.

**Methodology:** Pedro R. Lowenstein, Sebastien Motsch.

**Software:** Sara Jamous, Sebastien Motsch.

**Visualization:** Sara Jamous, Andrea Comba, Sebastien Motsch.

**Writing – original draft:** Sara Jamous, Pedro R. Lowenstein, Sebastien Motsch.

**Writing – review & editing:** Pedro R. Lowenstein, Sebastien Motsch.

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
