## [Decision Letter · Decision Letter 0]

24 Feb 2020

Dear Dr. Motsch,

Thank you very much for submitting your manuscript "Self-organization in brain tumors: how cell morphology and cell density influence glioma pattern formation" for consideration at PLOS Computational Biology. As with all papers reviewed by the journal, your manuscript was reviewed by members of the editorial board and by several independent reviewers. The reviewers appreciated the attention to an important topic. Based on the reviews, we are likely to accept this manuscript for publication, providing that you modify the manuscript according to the review recommendations.

Sincerely,

Philip K Maini

Associate Editor

PLOS Computational Biology

Feilim Mac Gabhann

Editor-in-Chief

PLOS Computational Biology

[LINK]

Reviewer's Responses to Questions

**Comments to the Authors:**

Reviewer #1: The authors propose a novel agent-based model (ABM) that simulates patterns of self-organization of elliptic/ellipsoid-shaped glioma cells in 2 & 3D. This is an exciting contribution and the authors are correct that studying the complex dynamics of such emergent flocks and (onco)streams – cells moving in the same/opposite directions - may potentially hold great promise for assisting in the development of innovative cytoreductive and cytotoxic therapeutic strategies, much needed for these extremely difficult to treat brain neoplasms.

Like any other model, this one has its limitations as e.g. the absence of biochemical factors reduces its levers to biomechanics, which is perfectly ok as long as one keeps this in mind when evaluating the results. For instance, the authors correctly emphasize that it is contrary to most concepts that an increase in cell density would lead to a reduction in flocks, and present this as an unexpected, noteworthy finding. Well – if cell-cell ‘avoidance’ is hard-coded in this ABM (pgs. 2 &3), wouldn’t one assume that cell ‘repulsion’ and/or change of cell directionality or ‘steering’ will become naturally more pronounced, particularly at higher N. The difficulty I am having with this is that real cancer cells are characterized by what is known as a loss of ‘contact inhibition’ … in other words, they exhibit the opposite of such ‘cell-cell avoidance’. It is precisely this feature that allows cells to form 2D-cell layer clusters in dishes and multicellular tumor spheroids in gels. In this context, the authors perhaps may find this paper an interesting read:

https://onlinelibrary.wiley.com/doi/abs/10.1046/j.1365-2184.2001.00202.x

For the sake of the argument, once the import of cell-cell avoidance is reversed and perhaps (in form of PDEs in a discrete-continuum extension of this work) one would introduce biochemical factors secreted by the agents/cells (autocrine, with paracrine impact) in response to microenvironmental triggers, cells at higher N [a non-explicit representation of either a replication capability of the agents (say depending on space-lattice availability in this biomechanical setting) or a dynamic Tumor growth term, N_t which are both missing here, say Gompertz or Logistic] would likely display more same-direction flock and less opposite-direction streaming behavior, particularly as N increases, which is realistic in a spatially confined area (3D tissue, skull); similarly, the impact of (different cell size/shape) ‘packing’ – an exciting extension of this work, correctly mentioned on pg. 13 - will likely be less pronounced in this cell-cell avoidance setting. Rather, one would think that securing cell outflow from a rapidly proliferating tumor surface is critical to manage the onset of central apoptosis and necrosis (see clinical MRIs with circular contrast enhancement) which in turn is to some degree a function of dwindling vascular support across the surface. In this 3/4D scenario, what would be interesting to study – with a model such as this one - is if directionality, steering angle & velocity/speed change as a function of the distance to the main tumor – with the hypothesis that the closer the individual cell is to the main tumor, in an area of metabolic constraint and reduced tissue carrying capacity, the more likely it is to follow others departing in the same direction (away from the tumor), with a straight angle at higher speed – while at some distance, in a nutrient-rich environment with less mechanical confinement through the growing main mass, sparsely seeded invasive cells could explore multi-directionality and angles with higher degrees of freedom to forage (– analogous to complex systems in social insects like ants). This would likely result in fractal like invasive patterns that we have seen in gels and that (probably reinforced by fractal-like neovascular architecture) have been noted for real tumor boundaries in the past (see e.g. works by Cross S.S.).

In summary, to put the claim of this work i.e. having clinical potential into perspective, I would like to encourage the authors to discuss key limitations of their current work, both in setup and results, in more detail.

Finally, as a minor item, in Figure 1B, it appears that the blue labels being used to denote ‘cells’ – but this is an H&E staining and convention is that it depicts cell ‘nuclei’ in blue, cell bodies in pink. Perhaps I misunderstood, but worth clarifying.

Reviewer #2: The authors study an important question in tumor biology in general and glioma in particular. They investigate the effects of cell shape and density on organization and direction of movement. The model is elegant and the results are interesting. The presentation and Figures are clear. I have minor suggestions/questions:

1. Did the author observe the existence of streams in experimental data? Figure 1 is static; though some cells are elongated, the direction of movement can not be determined.

2. The dynamic behavior of circular cells is not surprising as it is predetermined by the equations; see page 4, lines 106-109. I suggest mentioning this point in the discussion.

3. I suggest combining Figures 10-12 into a single Figure.

**Have all data underlying the figures and results presented in the manuscript been provided?**

Reviewer #1: Yes

Reviewer #2: Yes

PLOS authors have the option to publish the peer review history of their article (what does this mean?). If published, this will include your full peer review and any attached files.

Reviewer #1: Yes: Thomas S. Deisboeck, MD, MBA

Associate Professor of Radiology

Massachusetts General Hospital

Harvard Medical School

Reviewer #2: No

Figure Files:

Data Requirements:

Please note that, as a condition of publication, PLOS' data policy requires that you make available all data used to draw the conclusions outlined in your manuscript. Data must be deposited in an appropriate repository, included within the body of the manuscript, or uploaded as supporting information. This includes all numerical values that were used to generate graphs, histograms etc. For an example in PLOS Biology see here: http://www.plosbiology.org/article/info%3Adoi%2F10.1371%2Fjournal.pbio.1001908#s5.
---

## [Editor Report · Decision Letter 1]

19 Mar 2020

Dear Dr. Motsch,

We are pleased to inform you that your manuscript 'Self-organization in brain tumors: how cell morphology and cell density influence glioma pattern formation' has been provisionally accepted for publication in PLOS Computational Biology.

Best regards,

Philip K Maini

Associate Editor

PLOS Computational Biology

Feilim Mac Gabhann

Editor-in-Chief

PLOS Computational Biology

---

## [Editor Report · Acceptance letter]

27 Apr 2020

PCOMPBIOL-D-19-02178R1 

Self-organization in brain tumors: how cell morphology and cell density influence glioma pattern formation

Dear Dr Motsch,

I am pleased to inform you that your manuscript has been formally accepted for publication in PLOS Computational Biology. Your manuscript is now with our production department and you will be notified of the publication date in due course.

With kind regards,

Laura Mallard
